# Integrated Metabolite and Transcriptome Profiling-Mediated Gene Mining of *Sida cordifolia* Reveals Medicinally Important Genes

**DOI:** 10.3390/genes13101909

**Published:** 2022-10-20

**Authors:** Deepthi Padmanabhan, Purushothaman Natarajan, Senthilkumar Palanisamy

**Affiliations:** 1Department of Genetic Engineering, School of Bioengineering, SRM, Institute of Science and Technology, Kattankulathur 603203, India; 2Department of Biology, West Virginia State University, Institute, WV 25112-1000, USA

**Keywords:** gene expression, medicinal plants, acacetin, transcriptomics, biosynthesis, flavonoids

## Abstract

*Sida cordifolia* is a medicinal shrub that is conventionally used in the Indian system of medicine;however, the genes contributing to its medicinal properties have been minimally explored, thus limiting its application. High-throughputsequencing and Liquid Chromatography with tandem mass spectrometry(LC-MS/MS) technologies were applied to unravel the medicinally important bioactive compounds. As a result, transcriptomic sequencing generated more than 12 GB of clean data, and 187,215 transcripts were obtained by de novoassembly. These transcripts were broadly classified into 20 classes, based on the gene ontology classification, and 6551 unigenes were annotated using Kyoto Encyclopedia of Genes and Genomes (KEGG) database with more than 142 unigenes involved in the biosynthesis of secondary metabolites. LC-MS/MS analysis of three tissues of *Sida cordifolia* revealed that acacetin and procyanidin are some important metabolites identified thatcontribute to its medicinal value. Several key enzymes witha crucial role in phenylpropanoid and flavonoid biosynthetic pathways were identified, especially phenylalanine ammonia lyase, which might be an important rate-limiting enzyme. Real-Time Quantitative Reverse Transcription Polymerase chain reaction (qRT-PCR) analysis revealed enzymes, such as Phenylalanine ammonia lyase (PAL), Cinnamyl alcohol dehydrogenase 1 (CAD), Cinnamoyl-CoA reductase 1 (CF1) and Trans cinnamate 4-monooxygenase(TCM), which were predominantly expressed in root compared to leaf and stem tissue. The study provides a speculative insight for the screening of active metabolites and metabolic engineering in *Sida cordifolia.*

## 1. Introduction

Plants have been used as an immortal source of medicine forages. Globally, herbal medicines have played an important role in human health to treat chronic and acute conditions without any toxic effect. They are usually used as therapeutics to health conditions including diabetes mellitus, wounds, cancer, heart diseases, tuberculosis, hypertension, etc. These herbs are used in medicinal fields, as they are rich in bioactive phytocompounds, such as flavonoids, alkaloids, terpenoids, polyphenols, and tannins possessing various pharmacological properties [1]. A traditional sub shrub, *Sida cordifolia*, belonging to the family *Malvaceae*, is widely spread across countries like India, Brazil, and Africa and is extensively used in the ayurvedic system of medicine. It has laid the foundation to perform the transcriptomic approach to elucidate the various biosynthetic pathways responsible for its medicinal properties [2]. Some important constituents isolated from various extracts of *Sida cordifolia* are 1, 2, 3, 9-Terta hydropyrrolo[21-b]-quinazolin-3-yl-amine, ephedrine, vasicine, pseudo-ephedrine, vasicinone, hypaphorine, vasicinol, stigmasterol, and sterculic acid [3,4,5,6,7]. Pharmacological properties exhibited by different extracts of *Sida cordifolia* are antimicrobial, anti-inflammatory, analgesic, anti-ulcer, nephroprotective, anti-diabetic, hepatoprotective, anticancer, and central nervous system depressant activity [8]. The inevitable identification of genes responsible for the production of various secondary metabolites by *S.cordifolia* is achieved using the next-generation sequencing technology to unravel the novel transcripts from medicinal plants with respect to secondary metabolite biosynthesis and gene expression analysis [9]. The technology is employed to generate functional data for non-model plants and EST sequences for the annotation of genes and analysis of targeted discovery. The de novo sequencing prompts large volumes of information that can be used to identify various molecular markers, novel gene identification, or discovery and polymorphism [10,11,12,13]. Transcriptomics is the study of collection of all the transcripts available in the specific tissue andhas been used to study the structure and function of the gene at the molecular levels.The study of small molecular weight metabolites that exist in all cells to maintain their growth and function is referred to as metabolomics. The metabolite profiling reflects the overall biochemical and physiological conditions of the plant for the survival in that particular environment [14]. LC-MS analysis is preferable for evaluating large groups of secondary metabolites like phenolic compounds, flavonoids, and alkaloids [14]. Though the plant has been extensively used in herbal formulations, the biosynthetic pathways responsible for the synthesis of secondary metabolites that possess the pharmacological properties have yet to be unveiled. Thus, in this study, we aimed to annotate the transcripts, identify the putative transcripts that are involved in the major biosynthetic pathways attributing to the medicinal properties of the plant using transcriptomics, and analyze the different metabolites present in them by metabolomics.

## 2. Materials and Methods

### 2.1. Sample Collection and RNA Extraction

Root, stem, and leaf tissues were collected from a healthy plant near Potheri in Tamil Nadu, and identification was taxonomically done in SRMIST, IIISM. The RNA was extracted using TRizol^®^Reagent followed by phase-separation using Chloroform. After centrifugation at 12,000 rpm for 15 min at 4 °C. To the aqueous phase, 0.7 volume of isopropanol was added and centrifuged at 12,000 rpm for 10 min at 4 °C, followed by ethanol, which was to the pellet obtained from the previous step. Then, nuclease-free water was addedto the air-dried pellet. To remove the DNA contamination, the RNA was treated with DNase. A further purification of the total RNA was done using Qiagen RNeasy^®^MinElute clean-up kit. The integrity of RNA was analyzed using the Agilent Bioanalyzer with RIN value range of 7.4 to 9.2, a Thermo Scientific Nanodrop Lite spectrophotometer was used to check the quality of RNA, and the samples were visualized in the agarose gel electrophoresis [15,16].

### 2.2. RNA Library Preparation and Illumina Sequencing

Purified RNA was used for cDNA preparation. The poly A tail present in the mRNA was refined from the total RNA using magnetic beads attached to poly-T oligo, and a fragmentation buffer was used for fragmenting the mRNAs into smaller fragments. The first strand synthesis of these shorter fragments was carried out using Invitrogen’s Superscript II reverse transcriptase, and for the second strand synthesis, RNase H and DNA polymerase I were used. Single addition of dATP was done to the fragmented cDNA and then connected with adapters followed by a further selection of templates for PCR amplification. NextSeq 500 was used for sequencing, and the raw data in the FastQ format were obtained (Illumina Inc., San Diego, CA, USA) [15,16].

### 2.3. Assembly of Transcripts

The raw data obtained from sequencing were subjected to quality control and reads with (Phred score ≥ 30), were removed by the software FastQCv.0.11.9, and the adapter sequences were trimmed by Cutadapt software. TRINITY v2.14.0 de novo assembler was used for the re-construction of transcriptome data with three software modules: the Inchworm module assembles the transcripts by k-mer; the Chrysalis module clusters the contigs, thereby constructing a *de Bruijn* graph for the contigs; and the Butterfly module analyzes the contigs based on the graphs created and lists the isoforms. CD-HIT v4.8.1 program was used to reduce the sequence redundancy and increase the analysis performance [17,18].

### 2.4. Gene Ontology Classification and Functional Annotation

The de novo assembled transcripts of *Sida cordifolia* were collated against different databases including protein non-redundant database (nr) from NCBI with the BLASTX tool, and an E-value not greater than 1E-05 was considered as a significant match. Further Omics Box tools were used for gene ontology (GO) classification and annotation with enzyme codes (EC). The pathway mapping was done by KEGG Automated Annotation Server for retrieving the pathway maps of Kyoto Encyclopedia of Genes and Genomes (KEGG) [19].

### 2.5. SSR Identification

The microsatellites were identified using the Krait software, which offers an ultrafast and user-friendly graphic interface for investigating genome-wide microsatellites. The software also identifies VNTRS from a large genome size, locates the SSRs in the gene coding region, and statistically analyzes and plots the graphs [20]. Simple sequence repeats are short repeat DNA sequences with 1–6 bp in length, which have high polymorphism and can be used as a tool in genetic mapping, population genetics, and phylogenetic analysis [21].

### 2.6. Transcript Quantification

The SALMON quantification tool was used to quantify the transcript abundance from the sequenced data andcombines the dual-phase parallel interface algorithm. The tool quantifies the transcripts based on the GC content, quasi mapping was conducted for accuracy, and fast mapping of genes was conducted to study the expression levels [22].

### 2.7. Reverse Transcription PCR Validation

The validation assembled transcriptome data of *Sida cordifolia* from the root tissue by reverse transcription PCR was carried out. The full-length transcripts were found using the BLAST tool to find the complete transcript to design the primers. Then, the PRIMER BLAST tool from NCBI was used to design the primers, and in silico PCR was conducted, where Actin was considered as a housekeeping gene. The gene expression analysis was normalized using reverse transcription PCR, and gradient PCR was conductedto optimize the annealing temperatures from 52 °C to 60 °C. PCR amplification was carried out by the following parameters: 95 °C for 5 min, 95 °C for 30 s, 57–59 °C for 30 s, 72 °C for 30 s, and 72 °C for 5 min for 40 cycles.

### 2.8. Validation of Gene Expression Analysis

*S.cordifolia* transcriptome data were validated by quantitative real-time PCR with QuantStudio 5 (Thermo Scientific, Wilmington, DE, USA) PCR machine and the QuantiNovaSYBRGreen PCR (Qiagen Inc., GmbH, Germany) kit. Actin was used as an internal reference, and a negative control reaction was set up in all experiments where it was used. The relative gene expression levels were analyzed and calculated using the2^-ΔΔCt^ method, which represents the cycle threshold of the target gene with the housekeeping gene Actin. qRT-PCR analysis was carried out with three replicates [23].

### 2.9. Extraction of Metabolites from Sida cordifolia

The leaf, root, and stem tissues of *Sida cordifolia* were collected, dried using a microwave oven, and pulverized using an electric blender. Then, 10 g of powdered plant tissues were transferred to a Schott bottle, and 100 mL of 99.9% methanol was added to it and macerated for three days. The solvent was evaporated using a rotary evaporator. The samples were centrifuged at 3000 rpm, and the supernatant was transferred into a fresh tube and stored at room temperature until further analysis.

### 2.10. LC-MS/MS Analysis

The secondary metabolites were analyzed using Shimadzu LC-MS/MS-8040 (QQQ) (liquid chromatography–mass spectrometry, Triple Quadrupole, North America) with a scan range of 50–1000 *m*/*z* in both positive and negative modes. Mobile phase A: 5 mM ammonium formateand 0.1% formic acid in water; phase B: 5 mM ammonium formateand 0.1% formic acid in methanol with a flow rate 0.6 ml/min an isocratic A—20% and B—80% were used. The column Union was used at a temperature 40 °C with a flow rate of 500 μL/min. The data acquisition was done by the software LabSolutions™ LCMS, and the compounds were identified from free-source online tools, such as METLIN, PUBCHEM, and KEGG databases, along with some previously published articles.

## 3. Results

### 3.1. RNA Sequencing Analysis of Raw and Processed Data

The root transcriptome of *Sida cordifolia* was obtained with 59,484,771 raw reads. From the raw reads, adapter sequences were trimmed to reduce the redundancy, and 59,484,597 clean reads were obtained further with a GC content of 40.52%. The raw reads were submitted to NCBI Sequence Read Achieve (SRA) and an accession number PRJNA841821 was obtained. Overall, 187,215 transcripts were assembled using a de novo assembler TRINITY.The average length of bases was 1035.19, and the median contig length was 717. The contig N50 value was found to be 1622 (Table 1).

### 3.2. Functional Annotation of Unigenes

A similarity search using BLASTX was conductedagainst the nr database from NCBI; with a total of 187,215 unigenes assembled, a total of 54,375 unigenes were non-annotated and 132,840 unigenes were annotated (Figure 1). Due to inadequate information of the *Sida cordifolia* genome, some annotated genes were classified into predicted, uncharacterized, or hypothetical proteins.The results from BLASTX were transferred to OMICS BOX to annotate further. The similarity search of assembled transcripts showed high similarity with *Gossypium hirsutum, Gossypium raimondii, Gossypium arboretum, Duriozibethinus, Theobroma cacao, Herrania umbratica*, and *Quercus suber* and the least similarity with *Stipa magnifica* and *Stipa borysthenica* (Appendix A).

### 3.3. Functional Classification of Unigenes

Using the OMICS BOX software, the assembled unigenes were annotated onto GO classification into three different categories viz. molecular function, cellular process, and biological process.The unigenes were categorized into 60 subcategories. In the cellular component, 9467 unigenes were classified into 20 classes. This category includes intracellular anatomical structure with high unigenes in the organelle, cytoplasm, and membrane and the leastunigenes in the nucleoplasm, supramolecular complex, and external encapsulating structure. In the molecular function, a total of 19,593 unigenes were classified into 20 different classes with high unigenes in organic cyclic compound binding, heterocyclic compound binding, and ion binding and low unigenesin isomerase activity, protein-containing complex binding, and carbohydrate binding. In the biological process, 25,315 unigenes were grouped into 20 classes with the maximum number of unigenes in organic substance metabolic process, cellular metabolic process, and primary metabolic process and the minimum number in signal transduction, response to chemicals, and vesicle-mediated transport (Appendix A).

### 3.4. Biological Pathway Analysis

In total, 6551 unigenes were annotated using the KEGG database and assigned to 150 pathway maps. A total of 6409 unigenes were found to be annotated in metabolic pathways onto different categories, which include nucleotide metabolism (2539 unigenes), carbohydrate metabolism (347 unigenes), metabolism of co-factors and vitamins (1333 unigenes), energy metabolism (823 unigenes), biosynthesis of antibiotics (413 unigenes), xenobiotics biodegradation and metabolism (135 unigenes), lipid metabolism (267 unigenes), and amino acid metabolism (477 unigenes) (Appendix A). Further, 142 unigenes were classified into biosynthesis of secondary metabolites, which was further subdivided into different categories: sesquiterpenoid and triterpenoid biosynthesis (57 unigenes), terpenoid backbone biosynthesis (22 unigenes), ubiquinone and other terpenoid-quinone biosynthesis (26 unigenes), phenylpropanoid biosynthesis (11 unigenes), isoquinoline alkaloid biosynthesis (8 unigenes), and flavonoid biosynthesis (6 unigenes) (Figure 2).

### 3.5. Untargeted Metabolic Profiling of Sida cordifolia

Transcriptomic analysis of metabolic pathways and the validation of key metabolites require further confirmation by the identification of metabolites. The metabolomic analysis of samples was completed by the untargeted metabolomic method. There were a total of 298 different metabolites in leaf, stem, and root tissues of *S.cordifolia* in both the positive and negative mode. Flavonoids were the major group of secondary metabolites found in the metabolome analysis of *S.cordifolia* where Naringin, Cinnamic acid, Cinnamaldehyde, Caffeic acid, Kaempferol derivatives, and Quercetin were predominantly present in root, stem, and leaf tissues. Some tissue-specific metabolites, such as Rosmarinic acid, Caffeic acid and its derivatives, Kaempferol derivatives, Rutin, Quercitrin, and Gallic acid were identified in root tissue. Apigenin, Gallagic acid, Quercetin and its derivates, Caffeic acid and its derivatives, and Kaempferol and its derivatives were found in leaf tissue. In stem tissue, Ferulic acid, Gallic acid, Quinic acid derivative, and Malic acid were found (Table 2). In some metabolic pathways, such as, flavonoid biosynthesis, isoflavonoid biosynthesis, flavones and flavonol biosynthesis, phenylalanine, tyrosine and tryptophan biosynthesis, and ubiquonone biosynthesis, intermediate metabolites could be identified in the mass spectra. The results revealed that the transcriptomic and metabolic analysis of the transcripts that catalyzes the enzymes is consistent with their metabolites that belong to the pathway.

### 3.6. KEGG Enrichment Analysis of Transcripts Involved in Various Secondary Metabolite Biosynthesis

#### 3.6.1. TranscriptsInvolvedin Phenylpropanoid Biosynthesis Pathway

The compounds derived from this pathway are initiated by the amino acid phenylalanine which serves as a precursor. Precursor phenylalanine, which is converted into Cinnamoyl-CoA, p-Coumaryl-CoA, p-Coumarylquinic acid, Caffeoylquinic acid, Caffeoyl-CoA, Feruloyl-CoA and Sinapoyl-CA. Eleven genes were intricate in the biosynthesis pathway. The root transcriptome analysis revealed the presence of the genes Phenylalanine ammonia lyase (EC: 4.3.1.24, EC: 4.3.1.25), 4-Coumarate CoA ligase (EC: 6.2.1.12), trans-cinnamate 4-monooxygenase (EC: 1.14.14.91), Cinnamoyl-CoA reductase 1 (EC: 1.2.1.44), and Cinnamyl alcohol dehydrogenase (EC: 1.1.1.195) (Appendix A).

#### 3.6.2. Transcripts Involved in Flavonoid Biosynthesis Pathway

The precursor for flavonoid biosynthesis is phenylalanine. The pathway is elucidated by the conversion of phenylalanine to p-Coumaryl-CoA, and the rate-limiting enzyme is Chalcone synthase. The genes identified from the transcriptome data are trans-cinnamate 4-monooxygenase (EC: 1.14.14.91), Chalcone-flavonone isomerase (EC: 5.5.1.6), and naringenin, 2-oxoglutarate 3-dioxygenase (EC: 1.14.11.9) (Appendix A).

### 3.7. SSR Identification

A total of 187,215 transcripts were analyzed, and 36197 perfect SSRs were identified with a total length of 626,976 bp, and the average length of SSRs was 17.33. Relative abundance and density were found to be 186.77 and 3235.12, respectively. The most abundant repeat motifswere mono nucleotide (12,898;35.6%), followed by tri-nucleotide (12,732;35.2%), di-nucleotide (6868;19%), tetra-nucleotide (2526;6.98%), penta-nucleotide (653;1.8%), and hexa-nucleotide (520;1.44%). SSRs with 5tandem repeats were most common in *Sida cordifolia,* followed by 12 repeats and 7tandem repeats. Among mono-nucleotide repeat motifs, Awashighly abundant (35%).In tri-nucleotide repeats, the highest frequency was observed in AAG (11.51%) followed by AAT (4.36%) (Appendix A)

### 3.8. Gene Expression Analysis of Sida cordifolia

The qRT-PCR analysis was done to validate the transcriptome date and the expression analysis of the selected genes. The selected transcripts included Phenylalanine ammonia lyase (EC: 4.3.1.24, 4.3.1.25), Trans-cinnamate4-monooxygenase (EC: 1.14.14.91), Chalcone-flavonone isomerase (EC: 5.5.1.6), and Cinnamyl alcohol dehydrogenase 1(EC: 1.1.1.219) (Table 3). The primer sequences used for the analysis are depicted in the Appendix A. The analysis revealed varied expression patterns of the selected genes, where all the genes were upregulated in the root tissue rather than the stem and leaf. Actin was used as a housekeeping gene. The results obtained showed significant agreement with the transcriptome data (Figure 3).

## 4. Discussion

### 4.1. De Novo Assembly, Functional Classification, and Annotation

RNA sequencing is a high-throughput sequencing method thathas been an integral part of metabolome research in non-model species with relatively high rapidity. This sequencing technology is effective to study the annotation and to elucidate various transcripts responsible for the biosynthetic pathways, gene expression analysis, and distribution of secondary metabolites in different tissues [52].The Illumina sequencing of the root tissue of *Sida cordifolia* revealed 59,484,771 reads, which were further processed to remove the adapter and redundant sequences. The de novo assembler TRINITY was used to assemble the transcripts, and a total of 187,215 numbers of transcripts were identified. 

Functional annotation and classification comprise a vital step thatprovides us with the homology alignment. Gene ontology classification was done to annotate the transcripts into three major categories such as cellular function, molecular function, and biological process. In our study, out of 187,215 unigenes, a total of 54,375 unigenes were non-annotated, and 132,840 unigenes were annotated against different databases. *Sida cordifolia* showed high similarity with the genus *Gossypium* and the least similarity with *Stipa* genus. These results can be because of the scarcity of reference genomic resources for *Sida genus*.

### 4.2. KEGG Pathway Analysis and Identification of Candidate Genes Involved in Secondary Metabolite Biosynthesis

Biological pathway analysis against the KEGG database was conductedto identify the various pathways involved in the transcriptome data with a total of 150 pathway maps, out of which a higher number of transcripts were annotated into metabolism pathway than the secondary metabolite biosynthesis pathway maps. The phenylpropanoid biosynthesis pathway is the key biosynthetic pathway responsible for the production of various secondary metabolites. The pathway is initiated by catalyzing phenylalanine to cinnamate and ammonia by the enzyme Phenylalanine ammonialyase (PAL). PAL is the key regulatory enzyme involved in the pathway that is found ubiquitously in plants [53]. The enzyme serves as a precursor for the flavonoid and lignin biosynthetic pathways [54]. A total of 12 transcripts were identified that are responsible for synthesis of PAL gene. Trans-cinnamic acid is also a precursor for the flavonoid and biosynthetic pathways. The increased activity of the enzyme PAL in turn increases the production of phenylpropanoid products, which vary with different stress stimuli, developmental stage, and tissue and cell differentiation. The enzyme is delineated to be stimulated by infection, radiations, drastic change in temperature, or drought stress [55,56,57,58]. Cinnamoyl-CoA reductase 1 involved in the monolignol pathway engenders the conversion of p-coumaroyl-, feuloyl-, and sinapoyl-CoA to p-coumaraldehyde, coniferaldehyde, and sinapaldehyde [59,60].

Flavonoids are naturally occurring secondary metabolites that are rich in antioxidant, antibacterial, anti-inflammatory, antifungal, and antidiabetic activity and act as an anticancer agent. On the basis of a number of hydroxyl groups, flavonoids are grouped into flavones, flavonols, flavones, anthocyanidins, and isolflavones. Flavonoids are synthesized from the phenylpropanoid biosynthesis pathway with the key enzymes involved in them. The enzymes that are involved in the first step of flavonoids biosynthesis are 4-Coumaroyl-CoA ligase, 4-Coumarate-3-Hydroxylase, and Phenylalanine ammonialyase. The genes involved in flavonoid biosynthesis pathway are flavone synthase, Dihydroflavonol-4-Reductase, Chalcone synthase, and Chalcone isomerase [61], wherethe enzyme 4-coumarate-CoA ligase catalyzes the activation of 4-coumarate, which occurs in multiple isoenzymes forms and exhibits substrate affinity with the metabolic function. It has a pivotal role in the biosynthesis of secondary metabolites from general phenylpropanoid metabolism. Other secondary metabolite biosynthesis pathways identified are ubiquinone, triterpenoid, sesquiterpenoid and isoquinoline alkaloid biosynthesis pathways.

A flavonoid—quercetin—and its derivatives are extensively used due to their bioactive effects as they possess many pharmacological properties such as anti-arthritic, cardiovascular, anticancer, anti-Alzheimer’s, antimicrobial, and wound-healing effects[62,63,64]. Studies suggest that flavonoids are found to be potential inhibitors of coronaviruses against the SARS-CoV-2 infection by binding to the targets that promotes the replication and entry of viruses [65,66]. The KEGG pathway analysis revealed that the flavonoids and its derivatives are derived from the phenylpropanoid biosynthesis pathway from L-tyrosine with intermediate metabolites, such as Naringenin and Kaempferol. Phenolic compounds function by reducing the reactive oxygen species and stabbing the lipid peroxidation, thus acting as a potent antioxidant agent. They also play a vital role in prevention and treatment of chronic illnesses like neurodegenerative disorders. The integrated LC-MS/MS and transcriptomic analysis revealed that acacetin and procyanidin metabolites were some important secondary metabolites that could be responsible for the pharmacological properties exhibited by the plant, which is used to treat various inflammations and bronchodilator activity and has a cardiovascular protecting nature.

### 4.3. Simple Sequence Repeats Analysis

Molecular markers reveal the genetic relationship between the species as they are unaffected by the environment and have easy detection and high heritability [67,68]. These markers are widely used in plant improvement and genetic conservation [69]. A total of 36,197 SSRs were identified with highest number of mononucleotide motifs compared to di-, tri-, tetra-, penta-, and hexa-nucleotide motifs. SSRs with five tandem repeats were most common in *Sida cordifolia* with highly abundant A (35%) mono-nucleotide repeats and in tri-nucleotide repeats, and the highest frequency was observed in AAG (11.51%) followed by AAT (4.36%). The SSRs were discovered to help plant genetics for molecular reproduction as well as identification of candidate molecular markers for understanding the genetic variations in the *Sida* family.

## 5. Conclusions

To expedite molecular level studies in *Sida cordifolia* and the characterization of the root transcriptome to identify the unitranscripts responsible for the biosynthesis of secondary metabolites, since the root tissue of the plant contributesto several medicinal properties of the plant. The result suggests that flavonoids are the major secondary metabolites that are responsible for the medicinal properties exhibited by the plant. The assembled transcripts were validated using reverse transcription PCR method, and the expression levels of the key genes from the phenylpropanoid and flavonoid pathways wereobtainedusing qRT-PCR. The secondary metabolites identified from the metabolomic studies lead the way for understanding the molecular mechanism of pharmacological properties exhibited by *S.cordifolia*. This is the first study that integrates the molecular basis and metabolomics of this plant.

## Figures and Tables

**Figure 1 genes-13-01909-f001:**
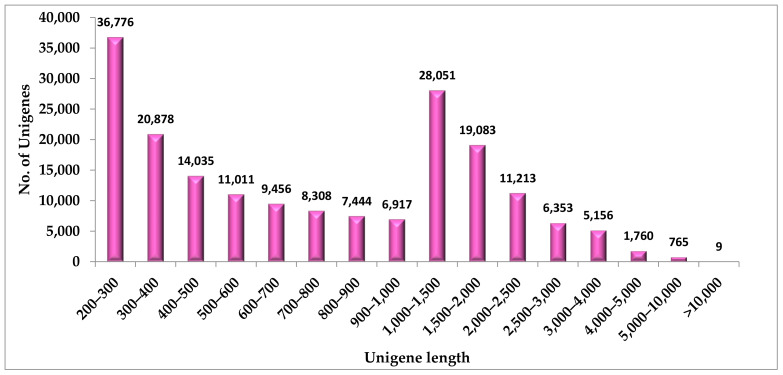
Unigene length distribution of *S.cordifolia* root transcriptome.

**Figure 2 genes-13-01909-f002:**
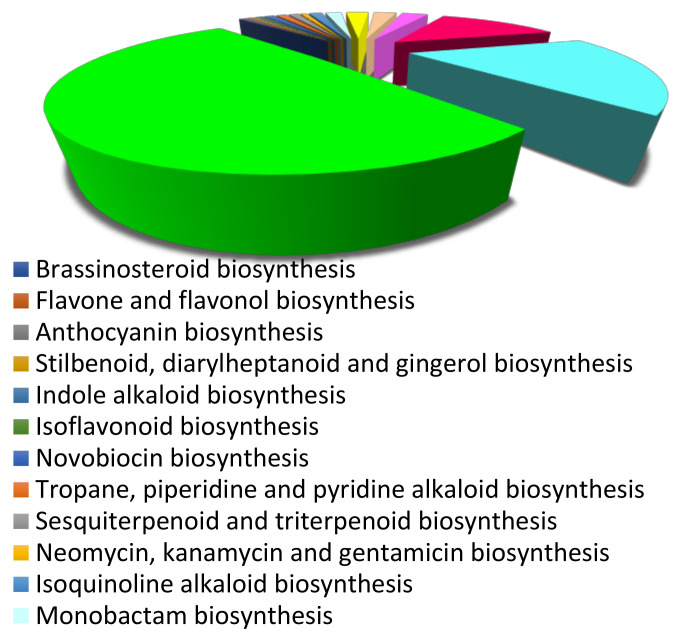
KEGG pathway analysis based on secondary metabolite biosynthesis in root transcriptome of *Sida cordifolia.*

**Figure 3 genes-13-01909-f003:**
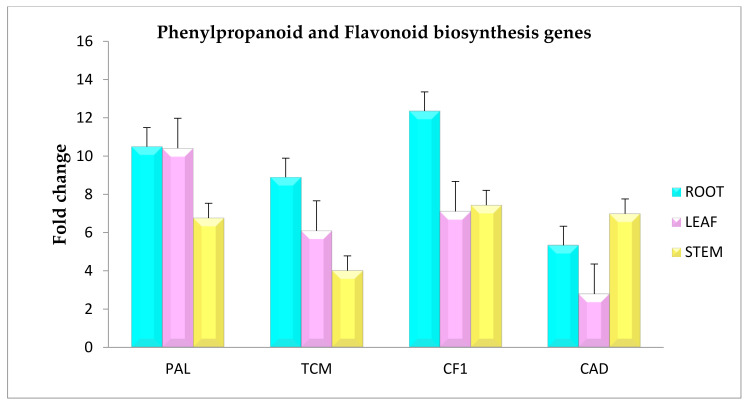
Validation of gene expression of phenylpropanoid and flavonoid biosynthesis genes by real-time PCR from *Sida cordifolia.*

**Table 1 genes-13-01909-t001:** Summary of Illumina paired-end sequencing and de novo assembly of *Sida cordifolia.*

Particulars	Numbers
Number of raw reads	59,484,771
Number of clean reads	59,484,597
No.of bases (after processing)	50,962,206
Mean Phred score	30
Total transcripts	187,215
Total length (bases)	193,803,049
Average length (bases)	1035.19
Median Contig length	717
GC content (%)	40.53
Contig N50 (Transcripts)	1622

**Table 2 genes-13-01909-t002:** Identification of various secondary metabolites from root, stem, and leaves tissue of *Sida cordifolia* by LC-MS/MS.

S.No	*m*/*z*	Compound Name	Pathway Involved in	Expressed in Root (R), Leaf (L), Stem (S)	Medicinal Properties	Reference
1	149.9, 148.9	Cinnamic acid	Flavonoid pathway	R, L, S	Antioxidant, antimicrobial, anticancer, neuroprotective, anti-inflammatroy, antidiabetic	[24]
2	315	Hydroxytyrosol 4-O-glucoside	Phenylpropanoid pathway	L	Anti-inflammatory, antioxidant, neuroprotective, immunomodulatory effects	[25]
3	133.0135, 115.009, 114.9900, 89.0200	Malic acid	TCA cycle	R, S	Antimicrobial activity	[26]
4	283.0617 177.0812 171.05 1510	Acacetin	Flavone pathway	R	Anticancer, anti-inflammatory	[27]
5	359.1372 285.0421 257.0316	Rosmarinic acid	Flavonoid pathway	R, S	Antimicrobial, immunomodulatory, antidiabetic, anti-allergic, anti-inflammatory, hepato and renal protectant	[28]
6	285.0396, 257.0512, 229.0516	Luteolin	Flavonoid pathway	R	Cardioprotective, antioxidant, antiviral, anti-inflammatory, antimicrobial, neuroprotective	[29]
7	489.1477 341.1138 179.0572 147.0313	Caffeic acid monohexoside derivative	Phenylpropanoid pathway	R, S	Antioxidant, anti-inflammatory, antineoplastic	[30]
8	327.1123 179.0571 147.0290	Caffeic acid derivative	Phenylpropanoid pathway	R, S, L	Antioxidant, anti-inflammatory, antineoplastic	[30]
9	577.2495 341.1110 179.0535	Caffeic acid-O- hexoside derivative	Phenylpropanoid pathway	R, S	Antioxidant, anti-inflammatory, antineoplastic	[30]
10	342	Caffeoyl glucose	Phenylpropanoid pathway	L	Antioxidant, anti-inflammatory, antineoplastic	[30]
11	311	Caffeoyl tartaric acid	Phenylpropanoid pathway	L	Antioxidant, anti-inflammatory, antineoplastic	[30]
12	179	Caffeic acid	Phenylpropanoid pathway	L, S	Antioxidant, anti-inflammatory, antineoplastic	[30]
13	567.228 405.1753 179.0515	Dihydrosinapyl caffeoyl hexoside	Phenylpropanoid pathway	R	Antitumor, neuroprotectant	[30]
14	461.1678 389.0100 385.1501 315.1100	Kaempferol-3-O-Glucuronide	Flavonoid pathway	R	Treats Alzheimer’s disease, Parkinson’s disease, ischemia stroke, epilepsy, major depressive disorder, anxiety disorders, neuropathic pain, and glioblastoma	[31]
15	417	Kaempferol-3-O-arabinoside	Flavone pathway	R, S	Treats Alzheimer’s disease, Parkinson’s disease, ischemia stroke, epilepsy, major depressive disorder, anxiety disorders, neuropathic pain, and glioblastoma	[31]
16	293	Kaempferol-3-O-(p-coumaroyl)-glucoside	Flavone pathway	S	Treats Alzheimer’s disease, Parkinson’s disease, ischemia stroke, epilepsy, major depressive disorder, anxiety disorders, neuropathic pain, and glioblastoma	[31]
17	301	Quercetin	Flavonoid pathway	R, L	Anticancer, antiviral, antiprotozoal, and antimicrobial effects;treatment of allergic, metabolic, and inflammatory disorders; eye and cardiovascular diseases; and arthritis	[32]
18	301 151 179	Quercetin hexoside	Flavonoid pathway	L	Anticancer, antiviral, antiprotozoal, and antimicrobial effects; treatment of allergic, metabolic, and inflammatory disorders; eye and cardiovascular diseases; and arthritis	[32]
19	610	Quercetin 3-O-glucoside	Flavonoid pathway	L	Anticancer, antiviral, antiprotozoal, and antimicrobial effects; treatment of allergic, metabolic, and inflammatory disorders; eye and cardiovascular diseases; and arthritis	[32]
20	610	Quercetin 3-O-rutinoside	Flavonoid pathway	L	Anticancer, antiviral, antiprotozoal, and antimicrobial effects; treatment of allergic, metabolic, and inflammatory disorders; eye and cardiovascular diseases; and arthritis	[32]
21	577.2699 417.1785	Procyanidin B2	Flavonoid pathway	R	Antioxidant, antibacterial, antitumor, anti-inflammatory, treat neurogenerative disorder	[33]
22	407, 425, 451, 289	Procyanidin dimer 1	Flavonoid pathway	L	Antioxidant, antibacterial, antitumor, anti-inflammatory, treat neurogenerative disorder	[33]
23	419.1364 337.1521 294.8212	Dihydroxymethoxy-glucopyranosylstilbene	Stilbene pathway	R	Anticancer, antitumor	[34]
24	429.17	Hecogenin	Saponin pathway	R	Anti-inflammatory, antioxidant, antifungal, hypotensive, anti-hyperalgesic, and anti-nociceptive effects	[35]
25	431.1631 413.3001 397.120, 201.0110	Kaempferol-3-O-rhamnoside	Flavonoid pathway	R	Anticancer	[36]
26	595.260, 448.181, 431.1930	Saponin cinnamoyl derivative	Saponin pathway	R	Fungicidal, antimicrobial, antiviral, anti-inflammatory, anticancer, antioxidant, and immunomodulatory effects	[37]
27	577.250, 431.2069	Kaempferol dirhamnoside	Flavonoid pathway	R	Antimicrobial, antioxidant	[38]
28	609.148, 449.160, 301.0412	Rutin	Flavonoid pathway	R	Antioxidant, anticonvulsant, anti-alzheimer, antidepressant, analgesic, antiarthritic	[39]
29	581.199, 419.350, 273.2141	Naringin	Flavonoid pathway	R, S	Antioxidant, anti-inflammatory, anticancer, anicarcenogenic, anti-ulcer	[39]
30	827	Naringin 6’-malonate	Flavonoid pathway	L	Antioxidant, anti-inflammatory, anticancer, anicarcenogenic, anti-ulcer	[39]
31	133.101, 131.08, 117.0665 115.050, 91.0212	Cinnamaldehyde	Phenylpropanoid pathway	R	Antifungal, antidiabetic,	[40]
32	197.117, 179.121, 135.110, 133.0211	Loliolide	Terpene pathway	R	Antioxidant, antifungal, antibacterial, antidiabetic, anticancer, antiviral, antituberculosis, anti-melanogenic, anti-inflammatory, and anti-aging	[41]
33	245, 179, 205	Catechin	Flavanol pathway	L	UV protectant, antimicrobial, antiallergenic antiinflammatory, antiviral, anticancer	[42]
34	353	Chlorogenic acid	Phenylpropanoid pathway	L, S	Antioxidant activity, antibacterial, hepatoprotective, cardioprotective, anti-inflammatory, antipyretic, neuroprotective, anti-obesity, antiviral, antimicrobial, anti-hypertension	[43]
35	355	Ferulic acid 4-O-glucoside	Flavonoid pathway	L	Vasodilator	[44]
36	339	3-p-Coumaroylquinic acid	Phenylpropanoid pathway	L	Antioxidant, antidiabetic, anticancer activity, antimicrobial, antiviral, anti-aging, protective, antinociceptive, and analgesic	[45]
37	191	Quinic acid derivative	Phenylpropanoid pathway	S	Antioxidant, antidiabetic, anticancer activity, antimicrobial, antiviral, anti-aging, protective, antinociceptive, and analgesic	[45]
38	315	Protocatechuic acid 4-O-glucoside	Tannin pathway	L	Antioxidant, anti-inflammatory, antihyperglyemic, antimicrobial,	[46]
39	465	Dihydromyricetin 3-O-rhamnoside	Flavonoid pathway	L	Antioxidant	[47]
40	273	3’,4’,7-Trihydroxyisoflavanone	Flavanone pathway	L	Inhibitor of UV radiation	[48]
41	605	Gallagic acid	Tannin pathway	L	Antioxidant, anti-inflamatory, antiplasmodial activity	[49]
42	593	Apigenin	Flavone pathway	L	Antioxidant, anti-inflammatory, anti-amyloidogenic, neuroprotective, and cognition-enhancing substance	[50]
43	193	Gallic acid	Flavonoid pathway	S	Antioxidant, anti-inflammatory, and antineoplastic	[51]

**Table 3 genes-13-01909-t003:** Major genes involved in phenylpropanoid and flavonoid biosynthesis pathways identified from *Sida cordifolia* root transcriptome.

Gene Name	EC Number	Unigene ID	Unigene Length	TPM Value	No. of Unigenes
Phenylalanine Ammonia Lyase (PAL)	4.3.1.24 4.3.1.25	TRINITY_DN21246_c0_g1_i1 TRINITY_DN21246_c0_g1_i2 TRINITY_DN21246_c2_g2_i2 TRINITY_DN21246_c2_g2_i3 TRINITY_DN22573_c0_g1_i1 TRINITY_DN22573_c0_g1_i2 TRINITY_DN21246_c0_g1_i1 TRINITY_DN21246_c0_g1_i2 TRINITY_DN21246_c2_g2_i2 TRINITY_DN21246_c2_g2_i3 TRINITY_DN22573_c0_g1_i1 TRINITY_DN22573_c0_g1_i2	358 313 2069 364 1135 983 358 313 2069 364 1135 983	518.921 288.181 170.628 123.988 57.6983 213.701 518.921 288.181 170.628 123.988 57.6983 213.701	12
Trans cinnamate 4-monooxygenase	1.14.14.91	TRINITY_DN19528_c1_g1_i1 TRINITY_DN19528_c1_g1_i7	2073 271	682.03 8.5076	2
Cinnamoyl-CoA reductase 1	1.2.1.44	TRINITY_DN16088_c0_g1_i9	3601	1.132281	1
Cinnamyl alcohol dehydrogenase 1	1.1.1.195	TRINITY_DN18521_c0_g1_i1 TRINITY_DN18521_c0_g1_i2 TRINITY_DN18521_c0_g1_i3 TRINITY_DN18521_c0_g2_i1 TRINITY_DN18521_c0_g1_i8 TRINITY_DN18521_c0_g1_i10	1921 1837 1450 1379 1724 2145	17.6824 32.1333 52.8276 4.49343 37.4843 44.9757	6
4-coumarate--CoA ligase 2	6.2.1.12	TRINITY_DN19412_c1_g1_i1	2201	16.54958	1

## Data Availability

The datasets used in the study was deposited in the SRA under the SRA accession number PRJNA841821.

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
