# Peer review of "Integrated Metabolite and Transcriptome Profiling-Mediated Gene Mining of Sida cordifolia Reveals Medicinally Important Genes"

_genes, 2022, doi:10.3390/genes13101909_

Round 1

Reviewer 1 Report

Dear editor,
Many thanks for send the manuscript ”Integrated metabolite and transcriptome profiling mediated gene mining of Sida cordifolia reveals medicinally important genes” for reviewing.
The manuscript first studies the root transcriptome and metabolism in Sida cordifolia. The work is important to reveal medicinally important genes. However, the manuscript needs to be revised first. The mainly problems were as follows:

1.    In the Introduction part, the author had few references on molecular study in this species, and had added the content to explain reason to integrate the two methods. And ref8.CNS first appeared should be explained.
2.    In the material and methods part, When and how old the species were collected by the author to extract RNA and the voucher information?
3.    Reverse transcription PCR validation part, under what standard, you selected the genes and designed the primers from the unigenes? And how to avoid the duplications?
4.    In gene identification parts, the author used the only four genes and does this in the same pathway related to the active compounds? It is better to selected genes in the same pathway related to the figured compounds.
5.    The author use the LC-MS to perform untargeted metabolite, what’s the MS condition and the test methods?
6.    The Figure 1 should be revised.
7.    The Table listed the compounds and should be put in as the attached files, the important compounds related to the genes should be listed.
8.    In the discussion, the author talk about more on transcriptome and less on metabolism, I think should combine the integration.

Author Response

The Editor 

Genes Journal

We thank you for the opportunity to revise the manuscript entitled “Integrated metabolite and transcriptome profiling mediated gene mining of Sida cordifolia reveals medicinally important genes” for publication in Genes Journal under the special issue “Phylogenetics, Genetics and Breeding of Medicinal Plants”. Following those suggestions we have made the revisions to the manuscript as outlined below. We look forward for your response and hope the revised version would be suitable for publication.

Thanking You

Senthilkumar Palanisamy

Reviewer 2 Report

The authors investigate the "Integrated metabolite and transcriptome profiling mediated gene mining of Sida cordifolia reveals medicinally important genes". The subject is interesting. However, the data seem only partly ok. In addition, the presentation and overall outlay of the manuscript require substantial revision.

1- What is the most important secondary substance of this plant?

2- Why is there no mention of it in the article?

3- Why are the genes involved in its biosynthetic pathway not investigated?

Please measure its amount and expression level using phytochemical and molecular methods and add it to the article.

4- Since it is mentioned in the introduction part of the article that compounds such as vasicinol, stigmasterol, and sterculic acid are the main compounds identified in the extract of this plant and have medicinal properties, why the genes of their biosynthetic pathway were not selected and the expression level of these genes was not measured.

5- What is your reason for selecting Phenylpropanoid and Flavonoid Biosynthesis pathways?

6- Why did you do the gene expression test for the root organ? 

If the amount of secondary compounds has been measured for leaves and stems.

Please do the amount of gene expression changes for other organs, compare them and add it to the article.

7- In section 2.9, what methanol percentage is used?

8- Add specific primers for real-time PCR tests.

9- In the Author Contributions section, the word "Dr" should be removed.

10- References should be formatted, for example, years. 

Author Response

We thank you for the opportunity to revise the manuscript entitled “Integrated metabolite and transcriptome profiling mediated gene mining of Sida cordifolia reveals medicinally important genes” for publication in Genes Journal under the special issue “Phylogenetics, Genetics and Breeding of Medicinal Plants”. Following those suggestions we have made the revisions to the manuscript as outlined below. We look forward for your response and hope the revised version would be suitable for publication.

Thanking You

Senthilkumar Palanisamy

Reviewer 3 Report

The present study shows the results of transcriptomic analysis of Sida cordifolia root tissue and it aimed to identify the genes involved in the biosynthesis of major bioactive compounds. For the first time, the Authors have provided insights into the molecular bases of secondary metabolite biosynthesis in Sida cordifolia roots. However, the manuscript in the presented form requires severe revision before publishing.

There are several major lapses in methodology description, results presentation, and analysis.

Please find attached all of my comments and suggestion.

Author Response

(The authors gave the same response as above.)

Round 2

Reviewer 1 Report

Thanks for sending the revised manuscript to me. The manuscript is important to Sida cordifolia study.

The many parts of the manuscript have been improved and it seems better. However, I still think that if the author add the mass condition in the supplementary, which would be better.

Author Response

Dear Editor

We thank you for the opportunity to revise the manuscript entitled “Integrated metabolite and transcriptome profiling mediated gene mining of Sida cordifolia reveals medicinally important genes” for publication in Genes Journal under the special issue “Phylogenetics, Genetics and Breeding of Medicinal Plants”. Following those suggestions we have made the revisions to the manuscript as outlined below. We look forward for your response and hope the revised version would be suitable for publication.

Thanking You

Senthilkumar Palanisamy

Response to reviewer 1’s comments

Thanks for sending the revised manuscript to me. The manuscript is important to Sida cordifolia study.

The many parts of the manuscript have been improved and it seems better. However, I still think that if the author add the mass condition in the supplementary, which would be better.

Thanks for your question. The mass conditions have been already mentioned in the materials and methods part of the manuscript. 

Reviewer 2 Report

1- As you mention in the abstract, the Acacetin and procyanidin compounds are some important metabolites identified, but, the authors just measured the Phenylpropanoid and flavonoid biosynthesis pathways that which are general pathways for other metabolites. Why did not investigate the expression genes involved in and the amounts of Acacetin and procyanidin? Since they are known as the most.

Please measure the amounts and expression level of them using phytochemical and molecular methods and then add them to the article.

2-References should be formatted, for example, years.

Author Response

Dear Editor

We thank you for the opportunity to revise the manuscript entitled “Integrated metabolite and transcriptome profiling mediated gene mining of Sida cordifolia reveals medicinally important genes” for publication in Genes Journal under the special issue “Phylogenetics, Genetics and Breeding of Medicinal Plants”. Following those suggestions we have made the revisions to the manuscript as outlined below. We look forward for your response and hope the revised version would be suitable for publication.

Thanking You

Senthilkumar Palanisamy

Response to reviewer 2’s comments

1- As you mention in the abstract, the Acacetin and procyanidin compounds are some important metabolites identified, but, the authors just measured the Phenylpropanoid and flavonoid biosynthesis pathways that which are general pathways for other metabolites. Why did not investigate the expression genes involved in and the amounts of Acacetin and procyanidin? Since they are known as the most.

Thanks for the question. Acacetin and procyanidin genes were not observed in the transcriptome data, so we couldn’t identify the genes responsible for its biosynthesis. Thus we have selected Phenylpropanoid biosynthesis pathway majorly as they serve as precursor for various other secondary metabolite biosynthesis for the validation of gene expression analysis.

Please measure the amounts and expression level of them using phytochemical and molecular methods and then add them to the article.

Thank you for your suggestion.   Acacetin and procyanidin biosynthesis genes were not observed in the transcriptome data. Thus, we have selected Phenylpropanoid biosynthesis pathway majorly as they serve as precursor for various other secondary metabolite biosynthesis for the validation of gene expression analysis.

2-References should be formatted, for example, years

The references are formatted as per suggestion.

Reviewer 3 Report

The Authors have made significant changes and improvements to the original manuscript; however, Results and Material, and methods are missing an explanation of how the Authors choose one of 12 PAL sequences for gene expression analysis. Furthermore, gene expression results lack statistical analysis, which is of utmost importance for quantitative research. The explanation of how the Authors normalized the results of gene expression is also missing from the Material and methods.

Author Response

Dear Editor

We thank you for the opportunity to revise the manuscript entitled “Integrated metabolite and transcriptome profiling mediated gene mining of Sida cordifolia reveals medicinally important genes” for publication in Genes Journal under the special issue “Phylogenetics, Genetics and Breeding of Medicinal Plants”. Following those suggestions we have made the revisions to the manuscript as outlined below. We look forward for your response and hope the revised version would be suitable for publication.

Thanking You

Senthilkumar Palanisamy

Response to reviewer 3’s comments

The Authors have made significant changes and improvements to the original manuscript; however, Results and Material, and methods are missing an explanation of how the Authors choose one of 12 PAL sequences for gene expression analysis. Furthermore, gene expression results lack statistical analysis, which is of utmost importance for quantitative research.

Thanks for the question. The full length transcripts were found using the BLAST tool to find the accurate complete transcript to design primers. Then the PRIMER BLAST tool from NCBI was used to design the primers and in silico PCR was done to test the designed primers. The qPCR experiments were performed with three biological and three technical replicates on the root, stem and leaf samples of Sida cordifolia. Error bars represent the standard errors to the legend in Figure 3.

The explanation of how the Authors normalized the results of gene expression is also missing from the Material and methods.

The gene expression analysis was normalized using reverse transcription PCR and gradient PCR was done to optimize the annealing temperatures from 52°C to 60°C.